# Effect of the Chain Structure of Self-Emulsifying Polyester Sizing Agent on ILSS of Carbon Fiber/Unsaturated Polyester Resin Composites

**DOI:** 10.3390/polym11091528

**Published:** 2019-09-19

**Authors:** Zhenyu Wang, Huijun Guo, Hang Zhou, Xinfeng Ouyang, Di Jiang, Jianhua Li, Qipeng Guo, Jun Tang, Chuncai Yang

**Affiliations:** 1Research School of High Performance Fiber and Composite, Jilin Institute of Chemical Technology, Jilin 132000, China; wang1994_zy@163.com (Z.W.); m15981210065@163.com (H.G.); zhouhang0126@126.com (H.Z.); joodi0204@126.com (D.J.); 13134419667@163.com (J.L.); 2China composites Group Co., Ltd., Beijing 100036, China; ouyangxf@ccgc.com.cn; 3Institute for Frontier Materials, Deakin University, Locked Bag 20000, Geelong, Victoria 3220, Australia; 4College of Chemistry, Jilin University, Changchun 130012, China

**Keywords:** carbon fiber, unsaturated polyester, sizing agent, ILSS

## Abstract

Herein, we report self-emulsifying anionic unsaturated polyester emulsions with different chain segments as novel sizing agents. The epoxy modified unsaturated polyester emulsions were synthesized via a self-emulsifying technique with no organic solvents. Emulsions were characterized by dynamic light scattering (DLS), Zeta potential, centrifuge, and cryo-scanning electron microscopy (Cryo-SEM). The results showed that the emulsions obtained were mono-dispersed nanospheres that had adequate colloidal stability. The maximum Zeta potential of the sizing agent is −52.88 mV. In addition, these emulsions were investigated as the sizing agents in order to improve interfacial adhesion between carbon fibers (CFs) and unsaturated polyester resin (UPR). Compared with the CFs sized with the commercial epoxy sizing agent, the interlaminar shear strength (ILSS) of CF/UPR composites from the CFs sized by these emulsion sizing agents with three different chain structures were enhanced by 25%, 29%, and 42%, respectively. The emulsion sizing agent composed of a flexible segment synthesized from adipic acid, neopentyl glycol, and maleic anhydride is most effective and can achieve the highest enhancement of the ILSS of CF/UPR composites.

## 1. Introduction

Carbon fiber (CF) reinforced composites have been widely used owing to their many outstanding advantages including light weight, excellent thermostability, and high specific strength [1,2]. The performance of composites usually depends on the matrix and fiber–matrix interface [3,4,5,6]. Excellent interfacial interaction ensures strong bonding between CFs and the matrix, which benefits the load transfer from the matrix to CFs [7,8]. On the other hand, the low surface energy and lack of functional groups of CFs results in the poor bonding between CFs and matrix [9,10]. In order to improve the adhesion, surface modifications of CFs have been conducted, including plasma treatment [11,12,13,14], chemical etching [15], chemical grafting [16], oxidation treatment [17], coating [18], and physical irradiation [19]. It has been demonstrated that the introduction of sizing agents does not cause secondary damage to the surface of fibers, and actually improves the wettability between fibers and resins [20,21] so that the interlaminar shear strength (ILSS) of composites can be enhanced [22,23].

CF/UPR composites have extensive application prospects in wind power generation, automotive components, and septic tanks [24]. Normally, epoxy sizing agents are widely used in CF composites in order to improve the wettability of carbon fiber [25], and the ILSS of CF/UPR composites are not strong enough when epoxy sizing agents are impregnated with CF so the application of CF/UPR is limited. To address this problem, Wu et al. introduced CNTs in a polyester sizing agent, remarkably enhancing the ILSS of CF/UPR composites [26]. However, unfortunately, in his experiment the sizing agent needs to be dissolved in acetone, which is toxic and dangerous chemical in carbon fiber manufacturing environment. Wang et al. synthesized an unsaturated sizing agent using 4′4-diaminodiphenyl methane and glycidyl methacrylate, effectively improving the interfacial adhesion of CF/VER composites [27], regrettably, the sizing agent was prepared by the phase inversion emulsification method, the storage stability of sizing agent is probably poor and the sized CF could be apt to moisture.

In this paper, three types of anionic unsaturated polyester sizing agent were prepared by a self-emulsifying technique, and no organic solvents were used in the whole synthetic process. Chemical structures, particle size polydispersity index, Zeta potential, centrifugal stability and the morphology of the sizing agent were characterized by Fourier transform infrared (FT-IR), nuclear magnetic resonance (^1^H-NMR), dynamic light scattering (DLS) centrifuge and cryo-scanning electron microscopy (Cryo-SEM). The effect of chain structure on the properties of the sizing agent was characterized by thermal gravimetric analysis (TGA) and differential scanning calorimetry (DSC). Surface morphologies and the surface energy of sized CF by different sizing agents were examined by scanning electron microscopy (SEM) and dynamic contact angle analysis (DCA), respectively. The interfacial properties of CF/UPR composites were measured systematically via interlaminar shear strength (ILSS). The results show that the ILSS of the CF/UPR composites from CFs sized by these novel sizing agents were greatly enhanced.

## 2. Materials and Methods

### 2.1. Materials

Phthalic anhydride (AR, 99%) and maleic anhydride (AR, 98.05%) were purchased from Shanghai Aladdin Biological Technology Co., Shanghai, China. Dipropylene glycol (AR, 99.5%), neopentyl glycol (AR, >96%), tetrabutylammonium bromide (AR, >99%), and cobalt naphthenate were supplied by Shanghai Macklin Biochemical Science and Technology Co., Shanghai, China. Adipic acid (AR, 98%) was purchased from Tianjin Guang Fu Science and Technology Development Co., Ltd., Tianjin, China. Bisphenol A-type epoxy resin, i.e., diglycidyl ether of bisphenol A (DGEBA), E-51, with epoxy equivalent 185–210, was supplied by Wuxi Blue-Star Petrochemical Co., Wuxi, China. Methyl tetrahydrophthalic anhydride was purchased from Guangdong Weng Jiang Chemical Reagents Co., Shaoguan, China. Triethylamine and tetrabutylammonium bromide (AR, >99.9%) were procured from Adama’s Reagent Co., Ltd., Shanghai, China. Acetone was obtained from Shanghai Ling Feng Chemical Reagent Co., Shanghai, China. 196 o-phenyl unsaturated polyester resins with viscosity of 800 Pa.S and 42 wt % of styrene was provided by Jilin 1000 Innovative Materials Co., Jilin, China. Methyl ethyl ketone peroxide (MEKP-9) was purchased from American Suzuki Co., Elyria, OH, USA. Polyacrylonitrile-based bare CF (BCF, unsized) and the commercial CF T300 (CCF, sized from epoxy sizing agent, sizing amount is 1.3 wt %) with the number of filaments carbon fiber bundles of 12000 were provided by Zhejiang Jinggong Carbon Fiber Co., Ltd., Shaoxing, China. Deionized water was produced in the laboratory. All materials were used as received without any further purification.

### 2.2. Synthesis of Self-Emulsifying Polyester Sizing Agents

Three types of self-emulsifying anionic polyester sizing agents, denoted as S1, S2, and S3, were obtained via the same procedures and from different monomers. The differences among S1, S2, and S3 originate from the synthesis of carboxyl-terminated unsaturated polyester resin by using different monomer as listed in Table 1.

Typically, for S1, phthalic anhydride (144.34 g, 0.975 mol) and dipropylene glycol (160.92 g, 1.199 mol) were placed into a 500 mL four-neck flask. The mixture was heated at 210~240 °C until the acid number of the mixture decreased to 5 mgKOH/g. Then, the mixture was cooled down to a temperature of 30 °C and maleic anhydride (29.4 g, 0.3 mol) was added. The temperature was increased again to 220 °C until the acid number dropped to 38 mgKOH/g and was left to cool down to 30 °C. Epoxy resinE-51(34.96 g) and TBAB (0.35 g, 0.001 mol) were then added and the reaction was continued up to 115 °C until the acid number was less than 5 mgKOH/g. Later, methyl tetrahydrophthalic anhydride (29.86 g, 0.18 mol) was added for further esterification reaction and the generation of carboxylic acid groups simultaneously. Finally, the S1 was obtained by adding TEA (18.18 g, 0.18 mol) and deionized water. The detailed synthetic route of S1 is given in the following Scheme 1. Compared to S1, S2 made by using adipic acid instead of phthalic anhydride, and S3 was synthesized in the same way as S2 but with neopentyl glycol to replace dipropylene glycol. The yields of S1, S2, and S3 are 80%, 79%, and 78%, respectively.

### 2.3. Sizing Process of Carbon Fibers

The BCF were sized continuously with the sizing agent in the apparatus shown in Scheme 2. The S1, S2 and S3 were diluted with deionized water to nonvolatile content: 3 wt %. The BCF were immersed into the immersion tank. Following sizing treatment, the carbon fiber bundles were removed from the tank and dried at 110 °C for 30 minutes in order to obtain sized carbon fiber bundles, denoted as SC1, SC2, and SC3, corresponding to S1, S2, and S3, respectively. The sizing amount of SC1, SC2, and SC3 are 1.25 wt %, 1.23 wt %, and 1.27 wt %, respectively.

### 2.4. Preparation of CF/UPR Composite

The unidirectional CF reinforced UPR composite materials were prepared by the hot-press molding method shown in Scheme 3. The unidirectional CFs sized by S1, S2, and S3 were impregnated with UPR system (1 wt % cobalt naphthenate and 1.2 wt % MEKP). Then the impregnated CF with UPR were laid flat in the mold, clamped by a C-type clamp and cured at 60 °C for 2 hours. The volume fraction of carbon fiber in CF/UPR composite is all 59%.

### 2.5. Characterization

S1, S2 and S3 were measured using an FT-IR Spectrometer (IR-100, Shimadzu, Tokyo, Japan) with KBr pellets. 1H-NMR spectra were recorded on a Bruker spectrometer (Bruker Avance 500, Bruker, Zurich, Switzerland) with CDCl3 as the solvent. The emulsion particle size and polydispersity index were measured by DLS (Omni, Brook, Holtsville, NY, USA) three times to give an average reading. Zeta potential measurements were also carried out on the DLS. In addition, the sizing agent morphology was examined with Cryo-SEM (JSM-7610F Plus, JEOL, Akishima, Japan) equipped with a cryo system (PP3010, Quorum, East Sussex, UK). Thermal gravimetric analysis (TGA, TG209F1, Netzsch, Exton, PA, USA) was used for the measurement of thermal degradation. The sample was under a N_2_ atmosphere with the temperature ranged from 30 °C to 500 °C at a heating rate of 10 °C/min. Differential scanning calorimetry (DSC, 204 F1, Netzsch, Exton, PA, USA) was used to examine the thermal properties at a rate of 10 °C/min under a N_2_ of 20 mL/min. The temperature ranges of the DSC tests were from −40 °C to 100 °C. The dynamic contact angle test method was applied to test the surface energy of CF. The dynamic advancing angle of CF with diiodomethane and water was tested firstly and then, through WORK theory and calculation, the surface energy, polar component and dispersion component of CF could be obtained by dynamic contact angle meter (DSA25, KRUSS, Hamburg, GRE). Based on GB/T 30969-2014, the ILSS tests were carried out on a Universal testing machine (AGS-X10KN, Shimadzu, Tokyo, Japan). The results were taken from an average value of at least seven specimens tested for each type of composites. The values of ILSS were calculated by the following equation:ILSS = 3*P*/4*bh*(1)
where *P* = maximum load observed during the test (N), *b* = measured specimen width (mm), and *h* = measured specimen thickness (mm).

## 3. Results and Discussion

### 3.1. Synthesis of Polyester Sizing Agent

A novel polyester sizing agent derived from distinct chain structures has been successfully synthesized with no organic solvents as shown in Scheme 1. At first, we synthesized carboxyl-terminated unsaturated polyester resin with different chain segments structures. Next, epoxy resins were added to take place reaction of epoxy ring with acid and to extend the chain. Subsequently, methyl tetrahydrophthalic anhydride was added to induce an esterification reaction selectively and to generate a hydrophilic carboxyl group at the same time, so that amphiphilic polymers were formed. Finally, triethylamine was used to neutralize the carboxyl group and the polyester sizing agent was obtained by adding water.

The chemical structures of S1, S2 and S3 were characterized by two spectroscopic techniques, FT-IR and ^1^H-NMR. Figure 1 displays the FT-IR spectra of S1, S2, and S3. All spectra exhibit a strong C=O vibration band at 1723 cm^−1^ and 1370 cm^−1^ confirming that unsaturated polyesters were synthesized. The νC–H stretch vibration was found near 2930 cm^−1^, confirming the presence of methylene groups. Furthermore, no characteristic bands of the epoxy ring near 915 cm^−1^ and 832 cm^−1^ were observed, illustrating that the epoxy resin is completely reactive. Moreover, the characteristic νO–H stretch vibration was not present at 3200~3700 cm^−1^, indicating that the hydroxyl group was esterified completely using methyl tetrahydrophthalic anhydride.

S1, S2, and S3 were further characterized by ^1^H-NMR to confirm the molecular structure. As shown in Figure 2, the ^1^H-NMR spectra of S1, S2, and S3 were similar to an extent. The unique peaks appeared in S1 were at 7.46 and 7.61 ppm were contributed to the benzene ring of phthalic anhydride. Due to adipic acid being used to replace phthalic anhydride in S2 and S3, no chemical shifts were observed at these points. Because of the dipropylene glycol used in S1 and S2, corresponding characteristic peaks can be observed at 3.4–5.2 ppm. In addition, the difference between S2 and S3 is the diols. The peaks in the data for S3 were at 3.87 and 4.00 contributed to neopentyl glycol. As a result of maleic anhydride was used in S1, S2 and S3, the chemical shifts can be seen at 6.84 and 7.1 ppm. The ^1^H-NMR and FT-IR analyses confirmed the formation of the S1, S2, and S3.

### 3.2. Emulsion Analyses of Sizing Agent

Colloidal stability is an important factor to evaluate the emulsions. The centrifugal stability and storage stability of S1, S2 and S3 were given in the Figure 3, which showed that S1, S2 and S3 have an adequate colloidal stability. Particle size and distribution of the sizing agent emulsion and Zeta potential has a great influence on the colloidal stability. Generally, particles in nano-size and the emulsion polydispersity index (PDI) of less than 0.25 will greatly enhance the colloidal stability [24,25]. In addition, if the value of Zeta potential is high (more than ±30 mV), the particles are prone to maintain stability due to the high electrostatic repulsion between particles [28].

The emulsion particle sizes of S1, S2, and S3 were showed in Figure 4, and the PDI of S1, S2, and S3 were 0.205, 0.103, and 0.157, respectively. From Figure 3, it can be identified that the particle size of S1, S2 and S3 are 135, 38, and 78 nm respectively, all of them are in nanometer scale. Additionally, Figure 5 shows that the Zeta potential of S1, S2, and S3 is −49.11, −50.97, and −52.88 mV, respectively. Furthermore, from Cryo-SEM micrographs (Figure 6), it can be further deduced that the sizing emulsion is mono-dispersed nanospheres.

### 3.3. Thermal Properties of Sizing Agents

The effect of the chain segmental structure on the thermal properties of the sizing agent was investigated by TGA as shown in Figure 7. There are two main chemical components which affected S1, S2 and S3 thermal stability. Compared S1 with S2, the chemical component difference is that S1 made from phthalic anhydride and S2 from adipic acid, the other chemicals are same. Because the π bond of the benzene ring attracts the electron cloud of phthalate ester of phthalic anhydride and lead to uneven electron density, so that phthalate ester has lower thermal stability than the ester bond of adipic acid, therefore S2 has much better thermal stability than S1. Similarly, S3 has better temperature resistance than S2 due to the component of neopentyl glycol for S3, that has a higher thermal stability than dipropylene glycol for S2.

Normally, at room temperature, CFs as reinforcing materials are usually weaved or braided in the manufacture of composite materials or textile processing. The CFs sized with a lower *Tg* of the sizing agent reduced rigidity and therefore, a lower *Tg* is more appropriate [29]. The *Tg* values of S1, S2, and S3 were 20.5, −26.9, and −22.5 °C, and were obtained from the DSC measurement which can be seen in Figure 8. It was identified that S1 had a highest *Tg* compared with S2 and S3 because of the use of phthalic anhydride increases the rigidity of the sizing agent. As a result, SC1 is more rigid than SC2 and SC3, which leads to bad solubility and wettability between SC1 and UPR. The adipic acid of soft chain segments was selected to synthesize S2 and S3, and their *Tg* was lower than S1. S3 is made from neopentyl glycol instead of dipropylene glycol, and as a result, the *Tg* of S3 is higher than that of S2. It has been shown in this work that the components of acid and alcohols have a significant influence on the *Tg* of the sizing agent. In conclusion, the chain structure with soft segments should be selected to lower *T*g of polyester sizing agent.

### 3.4. Surface Energy

In CF/UPR composites, the wettability between CF and resin directly determines the mechanical properties of the composites [30]. Wettability depends mainly on the surface energy of CF and the surface tension of the resin. The dynamic contact angles of SC1, SC2, and SC3 with water and diiodomethane were measured. According to Owens-Wendt-Rabel-Kaelble (OWRK) and Young’s equation, we can get the surface energy of sized CFs [31]. It can be seen from Figure 9 that the surface energy of sized CF by S1, S2, S3 and commercial epoxy sizing ranges from 40 to 45 mN/m. The S3 sized CF, in particular, shows the highest surface free energy (45.04 mN/m), and as a result, has produced the most favourable wetting conditions.

### 3.5. Surface Morphologies of Carbon Fibers

The surface morphologies of unsized CFs and CFs sized by S1, S2, and S3 were investigated by SEM. In Figure 10, image (a) is for unsized CFs, there are many narrow parallel grooves distributed along the longitudinal direction of the fiber. In contrast, surface morphologies of CFs sized by S1, S2 and S3 change remarkably. The surfaces of CFs were impregnated with a protective layer, which effectively changed the surface morphology of the fibers, and some grooves had been filled up, allowing the CF to bond with the resin firmly.

### 3.6. Interfacial Adhesion of UPR to CFs

The properties of carbon fiber-reinforced composites are greatly influenced by the interfacial bonding strength between CF and the resins [32]. In this research, the interfacial adhesion of composites has been measured by ILSS. The ILSS and load-displacement curves were shown in Figure 11a,b, S3 sized CF/UPR composite produced the greatest ILSS at 50.85 MPa, which is increased about 42% from the epoxy sizing emulsion sized CF/UPR composite. The ILSS of CF/UPR composites from CF sized by S1, S2, and S3 is much higher than from CF sized by the epoxy sizing emulsion. In order to increase the interaction between sized CF with UPR, unsaturated double bond (C=C) was introduced to the chain segment of the sizing agent, which will attend a crosslinking reaction with UPR as shown in Figure 12. However, the most commonly used epoxy sizing agent lacks reactive functional groups with UPR so that the ILSS of CF/UPR composites from CF sized by S2 is much higher than by commercial epoxy sizing agents even if the surface energy of sized CF is comparable. The ILSS of CF/UPR composite from CF sized by S1 is the lowest among S1, S2, and S3 because S1 is made from phthalic anhydride, leading to the highest *Tg* and most rigid SC1, which has a poor solubility and wettability between CF and resin. The ILSS of CF/UPR composite from CF sized by S3 is higher than from CF sized by S2 because S3 made from neopentyl glycol is more oleophylic and has a better solubility with UPR.

The composite fracture surfaces were examined by SEM, as presented in Figure 13. S3 sized CF/UPR composite exhibits continuous boundary between fibers and matrixes. Epoxy sizing emulsion sized CF/UPR composite shows some fibers cohering with only a few scattered pieces of UPR in the cross section (shown in Figure 13d), indicating weak interfacial adhesion.

## 4. Conclusions

In this research, three novel types of unsaturated polyester sizing agent (S1, S2, and S3) were synthesized without using organic solvents. The chemical structure of S1, S2, and S3 was confirmed by FT-IR and ^1^H-NMR. The experimental results show that S1, S2, and S3 are self-emulsifying, mono-dispersed nanospheres with an adequate colloidal stability. Thermal and mechanical studies indicated that the chain segmental structure of sizing agents have effects on the ILSS of the CF/UPR composites, and the sizing agents made from softer and lipophilic acid and diols can enhance the ILSS of CF/UPR composites. After the treatment with S3 sizing agent, the ILSS of the composites reached a maximum of 50.85 MPa—this value was 42% greater than the composites with the commercial epoxy sizing treated fibers.

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
