# Peer review of "Effect of the Chain Structure of Self-Emulsifying Polyester Sizing Agent on ILSS of Carbon Fiber/Unsaturated Polyester Resin Composites"

_polymers, 2019, doi:10.3390/polym11091528_

Round 1

Reviewer 1 Report

The author did not mention if there any previous studies on epoxy modified anionic unsaturated polyester sizing agents. If there are some, the author should state the difference between his research and others.

The author must mention the key properties (from datasheet) for the materials used in his research (as shown below), if it is not available, at least mention the product code number.

"Phthalic anhydride and maleic anhydride were purchased from Shanghai Xian Ding Biological 65 Technology Co., China. Dipropylene glycol, neopentyl glycol, tetrabutylammonium bromide, and 66 cobalt naphthenate were supplied by Shanghai Macklin Biochemical Science and Technology Co., 67 China. Adipic acid was purchased from Tianjin Guang Fu Science and Technology Development Co., 68 Ltd., Tianjin, China."

What is the volume fraction of carbon fibre?

ILSS results are attracting many inquiries; to gain confidence, the author must show recent literature for Epoxy sizing sized CF with UP, in section 3.6 ( not in the introduction). I am sure that you will find many works of literature have studied CF/UP.

The author should explain why the ILSS is of S2 is quite higher than epoxy sizing although the surface energy (an indicator of wettability, the main reason of the enhancement in this research) values are almost equal?!!

Reviewer 2 Report

The authors have produced a work concerning the synthesis of new sizing agents for the production of CF/UPR composites.

The use of sizing agents and the effect of their structure on the properties of the obtained composites is still an issue and the subject results very interesting.

Unfortunately the Authors focused their attention on the synthesis of the sizing agents and on their characterization: only ILLS measurements have been carried out on the composites obtained with these agents. As stated in the title, a more extensive discussion on the effects of the chain structure of the different sizing agents on the composites final properties should be provided. In particular the mechanical properties of the CF/UPR composites obtained with the different sizing agents should be fully characterized: the ILLS measurements are promising, but they are not enough.

I would suggest to publish it on Polymers after major revisions.

English should be revised: the language results sometimes incorrect. The references reported in the Introduction are often not pertinent and sometimes not accurate. E.g. reference 1 and 8 (line 35 and 38) deal with glass fiber composites even if the Authors are reporting about carbon fibers composites; reference 7 (line 38) should deal with the interfacial interaction between CF and matrix, but the paper is about PAN/PBA electrospun fibers; reference 9 (line 39) is not pertinent dealing with polyethylene glycol (PEG) hybrid materials (the publication year is wrong: the right one is 2007); reference 25 is not pertinent too; etc.

        Reference 21 lacks.

        Furthermore, in the Referee’s opinion, some references reported at the beginning of the Introduction are too specific (e.g. ref. 2, line 35) and should be replaced with some more general papers such as reviews.

Page 2, lines 45-46: “…can conserve energy [24]”. It is not clear what Authors mean with this sentence and the reference does not help in understanding it. Please, explain the concept in a suitable way, adding convenient references. In the Referee’s opinion, some discussion on the synthesis of the three sizing materials should be provided in the Result and discussion section. Furthermore their characterization should be discussed more in detail. In particular the Authors should discuss the different thermal decomposition stability of the sizing and providing the NMR peaks attribution. Moreover, the Authors do not provide any information on the stability of the obtained emulsion. Which is their storage stability? Did the Authors any evaluation of it? In Figure 1, line 154, the Authors highlight the bands at 1370 and 1251 cm-1 but no discussion is reported in the text. Please, modify the text or the figure. The Authors report a comparison of the surface energy of CFs sized with S1, S2, and S3 with respect to CFs sized with an epoxy sizing emulsion. Anyway, no details of this epoxy sizing emulsion (using as reference) are present: is it a commercial available sizing? Which one? Was this sizing (and its effect on the CFs/matrix interfacial adhesion) previously reported in the literature? In Figures 9 and 11 SEM micrographs are reported. For a better comparison, all the images should be reported with the same magnification. The Authors report ILSS measurements but they do not provide the number of specimens tested for each type of composites. In Figure 10 a and b the Authors report ILSS and Load-displacement curves of CF/UPR composites. For a better comprehension it is suggested to use the same colors for each sample in both figures. Conclusions, pag.11, line 242: “…with an adequate storage stability”. The Authors do not report any evaluation of the emulsion storage stability. Please, add a paragraph about this study. Conclusions are too generic and expected. E.g. it is well renown that the use of an aliphatic acid in place of an aromatic one will lower the Tg of the final material.

Round 2

Reviewer 1 Report

I have no comments at that stage, as I believe that the author has responded efficiently to the comments.

Reviewer 2 Report

The Authors revised and amended the paper to accommodate most of the reviewer comments. Anyway, I would suggest to publish it on Polymers only after minor revisions.

See the comments reported below.

The Authors should read and check all the text: English should be revised and several typo errors are present precluding the comprehension of the text. Some references are still not pertinent. E.g. reference 1 and 2 should deal with the advantages of carbon fibers reinforced composites (such as light weight, excellent thermostability and high specific strength, line 33); thus the Authors should refer to papers that report on the use of these materials for high performance structural materials. The cited papers are not a good choice. Reference 24 (25 in the first version of the paper) should report on CF/UPR composites applications, but it does not.The Authors have to check all the references and be sure they are pertinent with the text. The Authors were asked to provide some discussion on the synthesis of the three sizing materials in the Result and discussion section, but they did not. They should provide at least the yields for each obtained sizing agents in the Materials and Methods section. Line 152. “no characteristic bands of the epoxy RESIN near 915 cm-1”. In the Referee opinion RESIN should be replace with the word RING. In Figure 3 the Authors report the so-call centrifugal stability test of the emulsions. In the Referee’s opinion, particle sizes and distributions of the emulsion at rest at different time should be provided to demonstrate their stability. Line 243-244. “which is increased by 41.56% from…”. The percentage should be rounded.
